# Recreational Nitrous Oxide Use and Associated Factors among Health Profession Students in France

**DOI:** 10.3390/ijerph19095237

**Published:** 2022-04-26

**Authors:** Camille Inquimbert, Yoann Maitre, Estelle Moulis, Vincent Gremillet, Paul Tramini, Jean Valcarcel, Delphine Carayon

**Affiliations:** 1Department of Public Health, Faculty of Dental Medicine, University of Montpellier, 34000 Montpellier, France; paul.tramini@umontpellier.fr (P.T.); jean.valcarcel@umontpellier.fr (J.V.); 2Institut Desbrest d’Epidémiologie et de Santé Publique, IDESP UMR UA11 INSERM, Université de Montpellier, 34093 Montpellier, France; 3EA 2415, Aide à la Décision pour une Médecine Personnalisée, Université de Montpellier, 34093 Montpellier, France; maitreyoann@yahoo.fr; 4Department of Pediatric Dentistry, Faculty of Dental Medicine, University of Montpellier, 34000 Montpellier, France; estelle.moulis@umontpellier.fr (E.M.); vincent.gremillet@outlook.fr (V.G.); 5Department of Prosthodontics, Faculty of Dental Medicine, University of Montpellier, 34000 Montpellier, France; delphine.carayon@umontpellier.fr

**Keywords:** nitrous oxide, students, health profession, public health

## Abstract

The first aim of this study was to investigate the recreational use of nitrous oxide (N_2_O) among health profession students at Montpellier University (France). The second aim was to identify the factors associated with N_2_O use. All students in medicine, dentistry, pharmacy and midwifery of the Montpellier University were contacted by email to participate in the survey. The students answered directly online by filling out anonymously a questionnaire including demographic information and questions about N_2_O, illicit drugs and alcohol use. Ethical approval was granted by the ethics committee of the Montpellier University. The sample comprised 593 students (mean age = 22.3 ± 2.6 yr), with 68.6% of females. Lifetime N_2_O use was reported by 76.6% and frequent alcohol use by 30.5% of the respondents. The lifetime use of cannabis, ‘poppers’, cocaine, ecstasy and LSD was 26.8%, 54.6%, 9.6%, 10.1% and 2.0% respectively. In multivariate analysis, the substances significantly associated with lifetime N_2_O were alcohol drinking and ‘poppers’ use. With respect to this self-nominated sample, our results indicate that respondents who were alcohol drinkers, were poppers users, follow longer studies, divert medical products for recreational use or were members of a students’ corporation had higher odds of lifetime N_2_O use.

## 1. Introduction

Nitrous oxide (N_2_O) is used as an aesthetic in different medical domains. Since it was first synthetized in 1772 by Joseph Priestley, it has been subject to hedonistic consumption. It is also known as laughing gas, and it is well known by health students and practitioners; mixed with 50% oxygen in an equimolar mixture of oxygen and nitrous oxide (EMONO), it is used as an anesthetic in French hospitals, particularly in dental surgery. It has been shown to be efficient in reducing or alleviating patient anxiety during dental treatments [1].

The recreational use of N_2_O is now very popular, particularly among adolescents and the student community, and it has been associated with multiple toxic effects after inhalation [2,3,4]. As a psychoactive drug, some cases of N_2_O abuse have been reported, which were associated with sensorimotor peripheral neuropathy or hematological disorders, with different manifestations, such as loss of sensory perception, mostly in the hands and the feet of the abuser [5,6,7]. Other cases of destructive properties and associated symptoms have been described, the most frequent being paresthesia, unsteady gait, weakness, feelings of dissociation, delusions and emphysema, and vitamin B12 deficiency [8]. Accidental deaths associated with recreational use have been seldom reported in UK [9]. This substance is easily available, namely in whipped cream charging bottles (known as whippits), and its consumption has increased substantially in different countries [3,8]. For recreational use, the bulbs are mostly released into a balloon containing 4 L of gas under normobaric conditions [10]. At the same time, a global increase of other recreational drugs consumption has been observed, including alcohol, cannabis, ketamine, alkyl nitrites (‘poppers’), Methylenedioxymethamphetamine (MDMA: ecstasy), hallucinogenic mushroom, cocaine or lysergic acid diethylamide (LSD), sometimes associated with N_2_O [11,12,13,14,15,16].

In France, it seems that the health students also frequently use N_2_O [1], but no study has ever been conducted about its recreational use, and little is known about either these students’ characteristics or their behavioral profiles. On a public health approach, it seemed relevant to study patterns of drugs consumption: alcohol, cannabis, ketamine, poppers, ecstasy, hallucinogenic mushroom, cocaine or LSD, which are the most commonly described among students, and to better understand their relation with N_2_O recreational use. The first aim of this study was to investigate the recreational use of N_2_O among the students of the medicine, dentistry, pharmacy and midwifery universities in the town of Montpellier (France). The second aim was to identify the factors associated with the N_2_O consumption.

## 2. Materials and Methods

### 2.1. Study Site and Population

In 2021, the survey investigated undergraduate students registered at the Montpellier University, until the sixth year in the medicine, dentistry, pharmacy universities and before the end of the third year of midwifery school. The Montpellier University welcomes all health students from the center-south region of France. All students were contacted by email to participate to the survey: 653, 348, 551 and 240 students from medicine, dentistry, pharmacy and midwifery participated, respectively. Participant’ recruitment was performed using the internet, leading to a self-nominated sample. Participants were informed about the purpose and procedure of the survey. The e-mail addresses database was provided by the administrator of each university (medicine, dentistry, pharmacy and midwifery) who was officially informed of the survey.

### 2.2. Ethics

The protocol and study design were approved by ethics and regulatory agencies and were implemented in accordance with provisions of the Declaration of Helsinki. The appropriate Committee (Local Research Ethics Committee, Montpellier, France) approved the protocol (ID 2020-009-12). Students who agree to participate to the study had to sign the inform consent.

The informed consent form contained the name and affiliation of the investigator, a plain language description of the study, the approximate duration of the interview and the ethics committee approval.

### 2.3. Measurement Tool

The students answered directly online by filling out an anonymous questionnaire in a spreadsheet format via Google Forms. It comprised three pages of A4 format with a total of 35 items (single-item question), including demographic information and questions about N_2_O, illicit drugs (poppers, ecstasy, cocaine, ketamine, LSD) and alcohol use. Students who declared to be N_2_O users were asked subsequent questions about behavioural characteristics, such as dose and frequency of consumption, place of purchase, the use of other substances, and the sensations caused by N_2_O. Cigarette and/or cannabis smoking, coffee and alcohol drinking were assessed by four levels: ‘never’, ‘sometimes’, frequently’ and ‘daily’. It took approximately 10 min to complete the questionnaire.

Information was collected from all respondents to the survey, via fill-in forms that were completed anonymously in the database. The respondents were informed online that the process was completely anonymous. A password enabled the clinician (VG) in charge of the data treatment to access the database. A pilot study was initially implemented on a group of 15 dental students in order to test the feasibility and the good understanding of the questionnaire. The final questionnaire was available online for a two-month period.

### 2.4. Statistical Method

Descriptive statistics were expressed via means and standard deviations for quantitative variables. Categorical variables were summarized as frequency counts and percentages with a 95% confidence interval. The primary outcome was the proportion of students who declared use of N_2_O. Bivariate analyses allowed a test of the relationship between the use of N_2_O and other variables such as demographic or behavioral variables (Pearson chi-square or Fisher test for categorical variables). Variables with a *p*-value of less than 0.20 were then proposed for multivariate analysis, through a multivariate logistic regression model using an automatic stepwise selection. The fit of the model was determined using an ROC curve and the area under the curve (AUROC) statistic as well as the Hosmer-Lemeshow chi-square.

The power calculation was derived from the logistic regression model which tested the effects of different covariables with the consumption of N_2_O (yes/no). Based on the work of Peduzzi [17], the minimum number of subjects to include was calculated by: N = 10 k/p, with ‘k’ being the number of covariates in the model (k = 20) and ‘p’ being the estimated proportion of N_2_O users (*p* = 0.40). It was found that the sample size would be at least 500 subjects.

All analyses were performed with the statistical software Stata 16.1 (StataCorp, College Station, TX, USA) based on a two-sided type I error with an *alpha* level of 0.05.

## 3. Results

### 3.1. Student Characteristics

With 593 completed questionnaires, the global response rate was 33.1%. It was 27.4%, 51.7%, 30.3% and 27.9% of the medicine, dentistry, pharmacy and midwifery students, respectively. The mean age of the sample was 22.3 years (±2.6 yr), with almost 20% of the students being older than 25 years. The majority of respondents were female (68.6%): the proportion was 56.1%, 69.8%, 70.1% and 95.5% of the medicine, dentistry, pharmacy and midwifery students, respectively. 

The proportion of N_2_O users among the respondents was 76.6% (n = 454, 95% CI: 72.9–79.9%), a proportion of 80.0%, 78.3%, 66.5% and 71.6% of the medicine, dentistry, pharmacy and midwifery students, respectively. A small proportion (2.6%) admitted to have given up. The specific proportions, according to sociodemographic variables are displayed in Table 1.

### 3.2. Harmful Habits According to N_2_O Use

In the bivariate analysis, different characteristics were significantly associated with N_2_O use, including males, older students, those following medical or dental studies, or being a member of a students’ corporation. Sixty-five percent of the respondents declared that they have never smoked, while 14.0% admitted smoking more than 5 cigarettes a day and 28.2% never drank coffee. Frequent alcohol use was reported by 30.5% (n = 181, 95% CI: 26.8–34.4%) of the respondents, a proportion of 28.7%, 36.5%, 28.7% and 6.1% of the medicine, dentistry, pharmacy and midwifery students, respectively. The proportion of respondents who never/rarely drank alcohol versus frequently/very frequently was not significantly different according to gender (*p* = 0.23). Cannabis use was reported by 26.8% of the students, with 3.2% using it frequently or very frequently. About half of the respondents (54.6%) declared using poppers, while 9.6% declared being lifetime user of cocaine, 10.1% of ecstasy, 12.8% of MDMA and 7.1% of mushrooms. Two percent declared being lifetime LSD users, a proportion of 2%, 4%, 0% and 0% of the medicine, dentistry, pharmacy and midwifery students, respectively.

Among the N_2_O users, 96.9% reported drinking alcohol, while the same was reported by only 76.3% of N_2_O nonusers. The proportion of tobacco users was 40.1% and 15.1% in the N_2_O users and nonusers, respectively. Cannabis consumption was reported by 31.5% and 11.5% of the N_2_O users and nonusers, respectively. In the bivariate analysis, different harmful habits, such as tobacco, alcohol, or cannabis consumptions were significantly associated with N_2_O use (Table 2).

In the multivariate analysis, N_2_O users were more likely to be alcohol consumers (*p* = 0.0001, OR = 2.48 [1.51;4.09]). The proportion of N_2_O users was significantly higher among those who had ever diverted medicinal products for recreational use (*p* = 0.0001, OR = 5.19 [1.84;14.62]). Although coffee, tobacco and cannabis consumption were higher among the N_2_O users in the bivariate analysis, these associations were no longer significant in the multivariate analysis.

### 3.3. Drug Use According to N_2_O Use

When investigating at the consumption of other substances, it appeared that ‘poppers’ were the most consumed drug, 67.8% and 11.5% among N_2_O users and nonusers, respectively, reporting use (Table 3). This was also the only substance to be significantly associated with N_2_O lifetime use in the multivariate analysis (*p* = 0.0001, OR = 8.98 [4.74;17.02]). Ketamine and LSD were never consumed by N_2_O nonusers, while they were consumed by 4.2% and 2.6% of the N_2_O users, respectively. The last-month users (even occasionally) showed similar proportions to lifetime users. We observed that the N_2_O nonusers never consumed ketamine, LSD or cocaine in the last month, while these proportions were 1.5%, 0.7% and 4.2%, respectively, among N_2_O users.

The multivariate regression model exhibited good discrimination (AUROC = 0.908 [0.880;0.936]) (Figure 1) and model fit (Hosmer-Lemeshow chi2 = 5.80, *p* = 0.67; Akaike’s information criterion = 383 and Bayesian information criterion = 422).

More than half of the respondents (58.2%) were using from 3 to 10 ‘whippits’ during one party. Most of them purchased it from the internet or from the students’ corporation. Only 1% obtained it from dealers.

Nearly one quarter of N_2_O users were not aware of their abuse, and a small proportion declared having increased their inhaling dose during the first lockdown. Desired effects, such as euphoria, impaired vision, potentiator effect, or fun were significantly associated with the inhaled dose (Figure 2).

The majority of respondents (88.8%) would recommend N_2_O use. Figure 3 shows the relationship between the students’ recommendations and the N_2_O dose usually inhaled.

## 4. Discussion

This was the first study about N_2_O recreational use ever conducted in France. It highlights the high proportion of health students with N_2_O use experience. Among the respondents: 76.6% (95% CI: 72.9–79.9%) reported a history including N_2_O use. The highest proportion was found in medicine students (80.0%) and the lowest in pharmacy students (66.5%). We can reasonably assume that the proportion of students, and of young adults in general, using recreational N_2_O has increased in the last five years. Another study found a high frequency of recreational N_2_O use in first-year students [2]. One study of adolescents (age 14–18 years) reported a proportion of 15.6% respondents who declared having used N_2_O at least once in their life [4]. But when considering those who would definitely or maybe use N_2_O in the future, the estimated percentage increases to 35–40%. These authors found that recreational N_2_O use was associated with binge drinking (OR = 2.49) and cannabis use (OR = 1.98).

The global response rate was 33.1%. The majority of respondents were females, which is frequent when self-selected individuals participate in surveys [18]. Evidence suggests women are generally more interested in health topics and are more compliant with prevention or follow-up visits.

Our findings also suggest a significant association with alcohol abuse (OR = 2.48) but not with cannabis use (*p* = 0.88). Furthermore, the association with poppers use was highly significant (OR = 9). This may indicate a typical trend in students’ behavior and harmful habits. This is another reason for intervention to target risk behaviors in adolescents and young adults. Moreover, this survey targeted individuals who are future health professionals, so the educational factor should have positively influenced their awareness of addictive behaviors.

The profile of students using N_2_O corresponded to those who declared diverting medicinal products for recreational use (multiple logistic regression: OR = 5.19), those who were in higher years of study (OR = 3.46), or those who were member of a students’ corporation (OR = 10.43), and those who were users of alcohol (OR = 2.48) or poppers (OR = 8.98). No association was found between N_2_O use and gender. This result was also found in other studies [4,10]. Although N_2_O use was significantly associated with the type of studies in the bivariate analysis, it was no longer significant in the multivariate model. It seems that the students from different academic backgrounds could often meet together at students’ parties regardless of their medical specialty.

Another finding was that the image that the respondents have of N_2_O is relatively positive, since 88.8% recommended its use to others. This could be explained by an easy availability of N_2_O, from supermarkets or even hospitals (EMONO). There was an obvious lack of concern with side effects, and some authors recommend that the prevention of N_2_O should start in early adolescence [4]. These authors reported that adolescents who once had used N_2_O inhalants also tended to use or continue to use other substances, which could ultimately lead to drug abuse.

The respondents declared that students’ parties or private house parties were their highly preferred choice for inhaling N_2_O, with 86% consuming between 1 and 10 ‘whippits’ at a single event. Considering the adverse effects, 6.4% declared ever collapsed after inhaling N_2_O and 23.6% were witnesses of somebody collapsing after inhaling N_2_O. Only 12.5% reported no effect after inhaling N_2_O, and 5.4% were consuming N_2_O during the first lockdown. There was a significant association between the dose inhaled and the presence of adverse effects. Previous studies did find such an association [3].

The preferred N_2_O providers were the internet, the students’ corporation and friends, while 4% admitted it was fraudulently taken at the hospital. This finding agrees with those of other studies [3,19] which reported that this substance is easily available to everyone. The euphoric effect was the favorite effect desired by the respondents. Almost half of the students appreciated its short and relatively harmless effect. This is reliable with its use at students’ parties or festivals.

It is well documented that health profession students are concerned by the current general trend of alcohol drinking and other illicit drug use [13,15,20,21,22]. In the present study, the proportion of students consuming alcohol (frequently or daily) was 30.5%, which is comparable with other studies that found the use rate to be 27% [23] and 30% [13] among dental students in the UK.

The proportion of respondents using lifetime cannabis was 26.8%, which is in accordance with other studies of students finding rates 37.2% in the USA [22], 33% among vocational dental practitioners from the UK [24], and 35% among dental students in the UK [15]. The proportion of students using lifetime cocaine was 9.6%, which is close to the proportion found in other studies: 8.5% among vocational dental practitioners from the UK [24], 14.1% from the USA [22], 4.5% among dental undergraduates in the UK [25], and 6.4% among medical students in the UK [20]. LSD was consumed by 2% of the respondents, ecstasy by 10.1%, amphetamines by 12.8%, and magic mushrooms by 7.1%; these proportions are similar to those reported by other studies [13,20,25].

Considering the sensitive topic of this study, is was essential that anonymity could be respected throughout the process. Indeed, anonymity encourages the participants to respond, because personal information or even intimacy is preserved, and it increases the honesty of responses.

### Limitations of This Study

Firstly, this study was limited to a single university, and although this university is one of the largest in the Southern France, the external generalizability of the results is not ensured. Moreover, giving that the sample was self-nominating, this could lead to a volunteer effect, affecting the representability of N_2_O users in the whole population of medical students. It seems that individuals are more likely to participate in a survey if it concerns a subject of interest to them [18]. Therefore, the prevalence of recreational N_2_O use may be overestimated in the present study and is probably overestimated in the majority of reported studies based on a voluntary approach. However, reasonable general population inferences can be drawn from surveys of drug users employing purposive sampling that seek to include a large number of participants. It has been previously demonstrated that self-report studies are a valid and effective method for describing drug user profiles [12]. Since this study was originally initiated by the dental faculty, the highest response rate was found among dental students (51.7%). As another consequence of the volunteer effect, dental students seemed to be more concerned than the other health students.

Another constraint was the cross-sectional design of this study, which could not detect certain trends over time. However, longitudinal studies are difficult to implement among students, so repeated cross-sectional surveys would be useful to analyse N_2_O use over time.

We also found many associations between illegal substances among this sample of students and high levels of poly-drug use. Therefore, recall bias and confounding factors can be considered as significant issues.

## 5. Conclusions

In conclusion, with respect to this self-nominated sample, our results indicate that respondents who were alcohol drinkers, were poppers users, follow longer studies, divert medical products for recreational use or were members of a students’ corporation had higher odds of lifetime N_2_O use. This substance was generally used at students’ parties or at private house parties, with other illicit substances used at the same occasion. Euphoria was the main desired effect after using N_2_O, and the almost-constant perceived side effect was tingling or numbness. Nitrous oxide, particularly with its EMONO mixing, is still widely used in medical care centers. As N_2_O is a psychotropic substance, it may lead to some dependency and side effects among individuals who ask for more at each visit to medical/dental care facility. Its misuse nowadays concerns different socioeconomic levels, including health profession students and adolescents from all backgrounds. We also found that with the internet, this substance could widely and easily be obtained, and that among N_2_O users, one third would recommend it for recreational use. It would be useful to develop future research on the adverse effects of nitrous oxide, to highlight the potential harms of its recreational use. Further investigation will be necessary to uncover whether N_2_O use could be the first step to other drug use among health students. School-based interventions could be appropriate to promote healthy lifestyle, including information about the risk of N_2_O use. Preventive recommendations would also be addressed to early students, in which it becomes an increasingly popular drug. This could become a public health concern in the near future.

## Figures and Tables

**Figure 1 ijerph-19-05237-f001:**
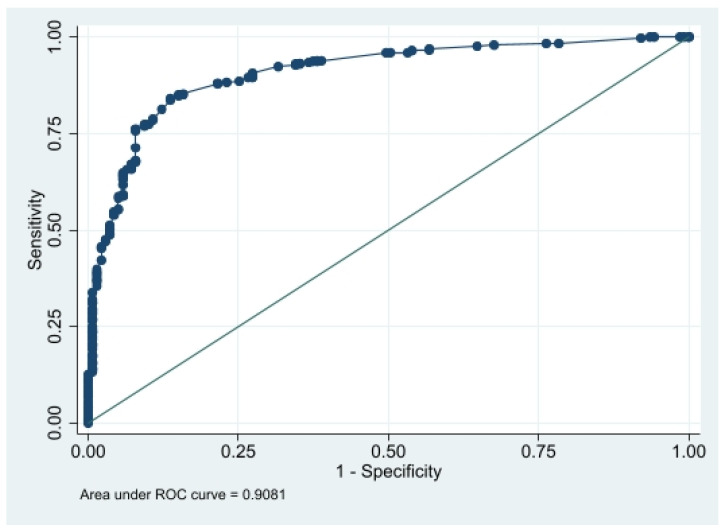
ROC curve after multiple logistic regression with a set of dependent variables including demographic variables (Table 1), harmful habits (Table 2) and other drugs (Table 3).

**Figure 2 ijerph-19-05237-f002:**
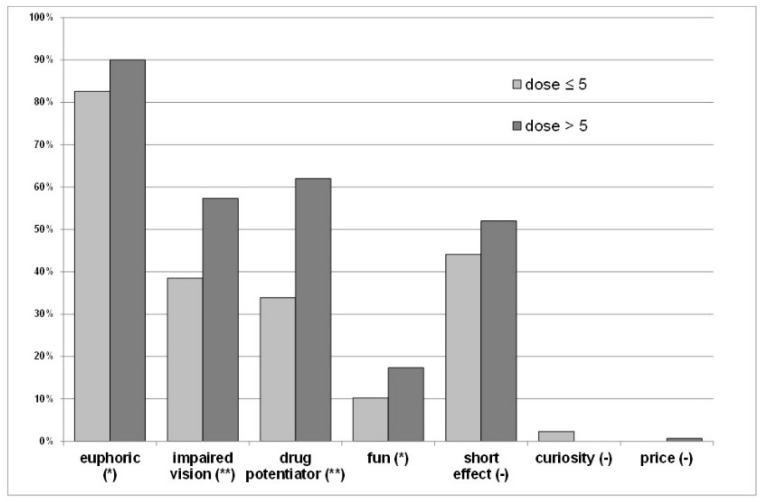
Prevalence of each needed effect in relation to the N_2_O dose. (*) *p* < 0.05 (**) *p* < 0.0001 (-) *p* ≥ 0.05.

**Figure 3 ijerph-19-05237-f003:**
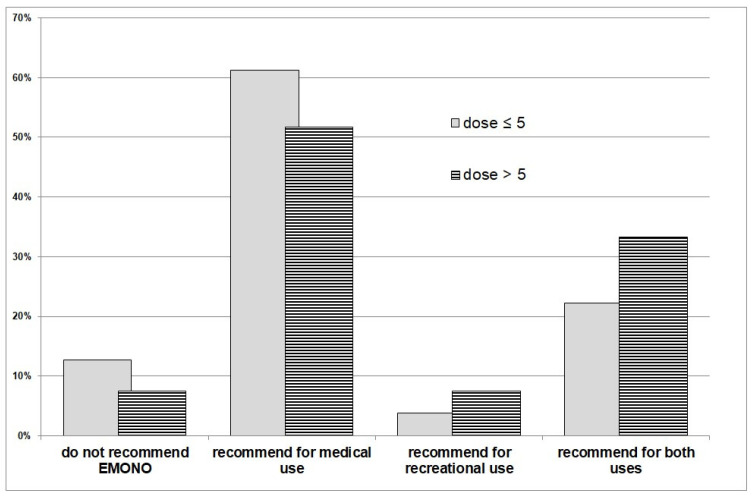
Student recommendations of EMONO according to N_2_O dose usually inhaled.

**Table 1 ijerph-19-05237-t001:** Sociodemographic characteristics of the students according to N_2_O use; comparisons between N_2_O users and non-users were made by bivariate test (chi-square) and multivariate test (multivariate logistic regression). (§) not in the final model after stepwise multiple logistic regression.

Variable	Never Used N_2_O		N_2_O Users		Bivariate Test	Multivariate Test
	n	%	n	%	*p*-Value	*p*-Value (OR)
Gender					0.04	0.89 (§)
Males	34	24.5	152	33.5		
Females	105	75.5	302	66.5		
Type of sudies					0.0001	0.31 (§)
Midwifery	19	13.7	48	10.6		
Medicine	25	18.0	154	33.9		
Odontology	39	28.1	141	31.1		
Pharmacy	56	40.3	111	24.4		
Age					0.0001	0.64 (§)
<21 yr	79	56.8	95	20.9		
21–22 yr	34	24.5	124	27.3		
23–24 yr	19	13.7	135	29.8		
>24 yr	7	5.0	100	22.0		
Years of study					0.0001	0.0001 (3.46)
1–2 yr	65	46.8	54	11.9		
3–4 yr	51	36.7	155	34.1		
3–4 yr	23	16.6	245	53.9		
Ever repeated year					0.16	0.07 (0.61)
No	46	33.1	180	39.7		
Yes	93	66.9	274	60.3		
Member of a students’ corporation					0.0001	0.002 (10.43)
No	137	98.6	345	76.0		
Yes	2	1.4	109	24.0		

**Table 2 ijerph-19-05237-t002:** Distribution of harmful habits according to N_2_O use; comparisons between N_2_O users and non-users were made by bivariate test (chi-square) and multivariate test (multivariate logistic regression). (*) not retained for the multiple logistic regression (*p* > 0.20). (§) not in the final model after stepwise multiple logistic regression.

Variable	Never Used N_2_O		N_2_O Users		Bivariate Test	Multivariate Test
	n	%	n	%	*p*-Value	*p*-Value (OR)
Divert medicinal products for recreational use					0.0001	0.002 (5.19)
No	134	96.4	338	74.4		
Yes	5	3.6	116	25.6		
Ever use antidepressants					0.26	(*)
No	128	92.1	403	88.8		
Yes	11	7.9	51	11.2		
Tobacco consumption					0.0001	0.88 (§)
Never	118	84.9	272	59.9		
Sometimes	11	7.9	109	24.0		
Frequently	4	2.9	24	5.3		
Daily	6	4.3	49	10.8		
Alcohol consumtion					0.0001	0.0001 (2.48)
Never	33	23.7	14	3.1		
Sometimes	93	66.9	272	59.9		
Frequently	13	9.4	162	35.7		
Daily	0	0	6	1.3		
Coffee consumtion					0.001	0.54(§)
Never	49	35.3	118	26.0		
Sometimes	38	27.3	85	18.7		
Frequently	21	15.1	65	14.3		
Daily	31	22.3	186	41.0		
Cannabis consumtion					0.0001	0.88 (§)
Never	123	88.5	311	68.5		
Sometimes	16	11.5	124	27.3		
Frequently	0	0	11	2.4		
Daily	0	0	8	1.8		

**Table 3 ijerph-19-05237-t003:** Distribution of other drugs use according to N_2_O use; comparisons between N_2_O users and non-users were made by bivariate test (chi-square) and multivariate test (multivariate logistic regression). (*) not retained for the multiple logistic regression (*p* > 0.20). (§) not in the final model after stepwise multiple logistic regression.

Variable	Never Used N_2_O		N_2_O Users		Bivariate Test	Multivariate Test
	n	%	n	%	*p*-Value	*p*-Value (OR)
Lifetime users:						
MDMA	3	2.2	73	16.1	0.0001	0.66
Ketamine	0	0	19	4.2	0.006	(§)
Mushrooms	5	3.6	37	8.2	0.07	0.09 (0.28)
LSD	0	0	12	2.6	0.04	(§)
Poppers	16	11.5	308	67.8	0.0001	0.0001 (8.98)
Cocaine	3	2.2	54	11.9	0.001	0.80
Last-month users (even occasionally):						
MDMA	1	0.7	29	6.4	0.001	(§)
Ketamine	0	0	7	1.5	0.14	(§)
Mushrooms	2	1.4	8	1.8	0.57	(*)
LSD	0	0	3	0.7	0.45	(*)
Poppers	4	2.9	132	29.1	0.0001	(§)
Cocaine	0	0	19	4.2	0.006	(§)

## Data Availability

The data presented in this study are available on request from the corresponding author.

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
