# Peer review of "Recreational Nitrous Oxide Use and Associated Factors among Health Profession Students in France"

_ijerph, 2022, doi:10.3390/ijerph19095237_

Round 1

Reviewer 1 Report

Dear Authors,
Congratulations for your research, it is simple and clear. It is well understood and well explained. Only, I find one flaw: limitations, prospective studies and practical applications are missing in discussion. Also add in methodology an explanatory figure of the research process to facilitate the understanding of the research.

Author Response

Thank you for your report. 

Methodology: add an explanatory figure of the research process

The research process was simple. The request to participate to the study concerned all the medical students: 653, 348, 551 and 240 students from medicine, dentistry, pharmacy and midwifery; and the response rates were 27.4% (n=179), 51.7% (n=180), 30.3% (n=167) and 27.9% (n=67) respectively.

Discussion: add limitations, prospective studies and practical applications.

A section about the limitations and constraints of the study has been added in the discussion. Prospective studies and practical applications have been added in the conclusion (the lack of concern with side effects and the need of preventive interventions).

Reviewer 2 Report

Dear editor

Thank you so much for the invitation to review the manuscript Recreational nitrous oxide use and associated factors among health profession students in France.

The manuscript is very well written, easy to read and the research design is appropriate.

Introduction, as expected sets the scene. the two aims are clear and the research design, as well as results, allow for their achievements.

Results well presented. 

Small correction versus, once it is Latin must be in italic in all text

Table legend must be more descriptive of the content. Some suggestions for figure legend.

Discussion very good and debating the obtained results. 

Missed the constraints of the study.

Conclusion - good.

The manuscript could be supported by a higher number of references. Some are old.

Author Response

Thank you for your report. 

Latin must be in italic in all text.

Changes have been made accordingly.

Table legends must be more descriptive of the content. Same suggestions for figure legends.

Table legends and figure legends have been modified and described more precisely.

Missed the constraints of the study.

A section about the limitations and constraints of the study has been added in the discussion.

The manuscript could be supported by a higher number of references. Some are old.

Only few references concerned the recreational use of N2O by health students and there is no such French study so far. A recent update gave very poor results, showing only distant relationship with the actual topic.

Reviewer 3 Report

Thank you for the opportunity to review this study entitled “Recreational nitrous oxide use and associated factors among health profession students in France” (ijerph-1672841).

The manuscript presented and exploration concerning the recreational use of the Nitrous oxide, also exploring its associated factors. A sample 593 students was involved in the study.

In my opinion the research topic is relevant, and the study is interesting. However, there are some issues that need to be addressed before the paper will be suitable for publication.

  • Abstract: Starting a sentence with "to investigate" in this way, without clarifying that the objective of the paper is being made explicit, confuses the first part of the abstract.
  • In the abstract, the information about the sample should be deepened (N? Mean age and SD? Percentage of men and women?) to provide a clear picture of what will be presented in the paper.
  • Please remove the headings from the abstract, in line with the journal's guidelines.
  • The introduction should be framed to better frame the phenomenon and justify the research objectives (e.g., why are certain factors analyzed to identify the ones associated with the N2O consumption?)
  • "The aim of this study was to investigate the recreational use of N2O among the students of the medicine, dentistry, pharmacy and midwifery universities in the town of Montpellier (France)." Please make it clear that this is the FIRST aim of the study.
  • More information on recruiting should be provided. For example, the authors declare “All students were contacted by email”. Where were the emails taken? On which database?
  • More information on administration should also be provided. I understand that the recruitment took place online, but what about the administration? The authors talk about an A4 format, but later they talk about an online questionnaire. What platform was it uploaded to? How long was the average duration for completing the questionnaire?
  • Has no statistical analysis been carried out on the tools used (e.g., alpha)? Were the dimensions analyzed evaluated with single-item questions? This is an important limitation of the study.
  • Please add a clear section of limitations and suggestions for future research.

Author Response

Thank you for your report. 

Abstract: the sentence starting with “to investigate” must be clarified.

This sentence has been modified in the abstract.

The information about the sample should be deepened (N, mean age and sd, percentage of men and women).

We added all these global results in the abstract.

Remove the headings of the abstract.

The headings have been removed.

Introduction: the introduction should be framed to justify the research objectives (factor analyzed for the association with N2O consumption).

The introduction has been partly modified accordingly.

Add the “first” aim of the study was too…

This sentence has been modified in the introduction.

Methods: more information on recruiting should be provided. Where were the emails taken? On which database? More information on administration should also be provided.

This information has been added in the ‘Methods’ section.

Precise why the A4 format.

We did not mean a sheet of A4 paper, but A4 size on the computer (attached ‘Word’ file).

What platform was it uploaded to? How long was the average duration for completing the questionnaire?

All the data from the questionnaires were processed in a spreadsheet format via Google Forms. This was added in the ‘Methods’.

It took approximately 10 minutes to fill out the questionnaire (link to URL: https://docs.google.com/forms /d/e/1FAIpQLSeD_xVTArsZoidC-3NachPaf3QnJ1JEdZKTS2WxgKrtiGpWUw/viewform?usp=sf_link).

Precise the alpha error in the statistical analysis. Were single-item questions?

Stastistical analyses were based on a two-sided type I error with an alpha level of 0.05 (‘Methods’ section).

All the questions were single-item. It was added in the ‘Methods’.

Discussion: add a clear section of limitations and suggestions for future research.

A section about the limitations and constraints of the study has been added in the discussion and suggestions for future research in the conclusion.

Round 2

Reviewer 3 Report

The authors responded adequately to all my comments and effectively resolved every issue I have raised. Thank you